# Ganglioside GD3 Regulates Inflammation and Epithelial-to-Mesenchymal Transition in Human Nasal Epithelial Cells

**DOI:** 10.3390/ijms25074054

**Published:** 2024-04-05

**Authors:** Ji Hyeon Hwang, Jae-Sung Ryu, Jin Ok Yu, Young-Kug Choo, Jaeku Kang, Jong-Yeup Kim

**Affiliations:** 1Department of Otorhinolaryngology-Head and Neck Surgery, College of Medicine, Konyang University Hospital, Daejeon 35365, Republic of Korea; wgdrww5@gmail.com (J.H.H.); jsryu@kbiohealth.kr (J.-S.R.); 2Department of Pharmacology, College of Medicine, Konyang University, Daejeon 35365, Republic of Korea; 3Department of Biological Science, College of Natural Sciences, Wonkwang University, Iksan 54538, Republic of Korea; yjo9703@naver.com (J.O.Y.); ykchoo@wku.ac.kr (Y.-K.C.); 4Institute for Glycoscience, Wonkwang University, Iksan 54538, Republic of Korea; 5Priority Research Center, Myunggok Medical Research Institute, College of Medicine, Konyang University, Daejeon 35365, Republic of Korea

**Keywords:** chronic sinusitis with nasal polyps (CRSwNP), ganglioside GD3, inflammation, epithelial-to-mesenchymal transition (EMT)

## Abstract

Chronic sinusitis with nasal polyps (CRSwNP) is one of the most common chronic inflammatory diseases, and involves tissue remodeling. One of the key mechanisms of tissue remodeling is the epithelial-mesenchymal transition (EMT), which also represents one of the pathophysiological processes of CRS observed in CRSwNP tissues. To date, many transcription factors and forms of extracellular stimulation have been found to regulate the EMT process. However, it is not known whether gangliosides, which are the central molecules of plasma membranes, involved in regulating signal transmission pathways, are involved in the EMT process. Therefore, we aimed to determine the role of gangliosides in the EMT process. First, we confirmed that N-cadherin, which is a known mesenchymal marker, and ganglioside GD3 were specifically expressed in CRSwNP_NP tissues. Subsequently, we investigated whether the administration of TNF-α to human nasal epithelial cells (hNECs) resulted in the upregulation of ganglioside GD3 and its synthesizing enzyme, ST8 alpha-N-acetyl-neuraminide alpha-2,8-sialytransferase 1 (ST8Sia1), and the consequently promoted inflammatory processes. Additionally, the expression of N-cadherin, Zinc finger protein SNAI2 (SLUG), and matrix metallopeptidase 9 (MMP-9) were elevated, but that of E-cadherin, which is known to be epithelial, was reduced. Moreover, the inhibition of ganglioside GD3 expression by the siRNA or exogenous treatment of neuraminidase 3 (NEU 3) led to the suppression of inflammation and EMT. These results suggest that gangliosides may play an important role in prevention and therapy for inflammation and EMT.

## 1. Introduction

Chronic sinusitis (CRS) is a prevalent inflammatory disorder affecting the mucosa of the paranasal sinuses. CRS is recognized for eliciting a protracted inflammatory response in the mucous membrane of the paranasal sinuses, consequently exerting a considerable influence on the quality of life of affected individuals [1,2]. CRS is typically categorized into two phenotypic subtypes: chronic sinusitis without nasal polyps (CRSsNP) and chronic sinusitis with nasal polyps (CRSwNP) [3]. Nasal polyps (NPs), benign tumors within the sinonasal region, represent abnormal proliferations located within the nasal cavity [4]. NP formation generally results from persistent inflammation in the nasal cavity, indicating a burden of disease as a recalcitrant and recurrent clinical course. Thus, the complexity of CRS leads to challenges in determining its etiology and developing treatments [2,3].

Tissue remodeling is a typical response to chronic inflammation and induces alterations in tissue structure. The epithelial-to-mesenchymal transition (EMT) is one of the crucial mechanisms involved in tissue remodeling [4,5,6]. EMT is a cellular process in which epithelial cells acquire mesenchymal properties and exhibit reduced cell–cell interactions and apicobasal polarity [7,8]. EMT occurs during embryogenesis, tissue repair, cell migration, fibrosis, and tumor invasion. It has been observed in CRS and NP tissues and is thought to be a pathophysiological process in CRS [9]. EMT is induced by several signaling pathways and hypoxia, such as transforming growth factor-β1, Wnt, TNF-α, hepatocyte growth factor, and Notch signaling [6,8,10,11,12]. Interestingly, CRSwNP is closely associated with tissue remodeling [13,14].

Gangliosides, complex glycosphingolipids characterized by the presence of one or more sialic acid residues, serve as prominent constituents within cytoplasmic membranes [15]. Various cell types exhibit a diverse range of gangliosides which participate in various biological processes, including apoptosis, cell proliferation and differentiation, cell-surface interactions, and transmembrane signaling [15,16,17,18,19]. Numerous studies have confirmed that the expression pattern and abundance of gangliosides are subject to developmental and cell type-specific regulatory mechanisms [20,21]. Notably, gangliosides play a key role in defining EMT-associated phenotypes [22,23,24,25,26]. Changes in glycosylation induced by chronic inflammation are associated with the development of several diseases [25,27,28]. However, it remains unknown whether gangliosides are involved in the EMT process.

In this study, we investigated the expression of EMT markers and gangliosides in patients with CRSwNP. Further, we analyzed the inflammatory response, EMT marker expression, and ganglioside expression in TNF-α-treated cells.

## 2. Results

### 2.1. Analysis of EMT Markers and Ganglioside Expression in Patients with CRS

Sinonasal tissue samples were collected from patients with CRSsNPs (UP tissues) and CRSwNPs (NP and UP tissues) as well as from control subjects (UP tissues) to measure EMT marker and ganglioside expression levels. Immunohistochemical analysis showed that the expression of the epithelial marker E-cadherin was higher in the UP tissues than that in the NP tissues. However, the mesenchymal marker, N-cadherin, was highly expressed in the NP tissues alone (Figure 1A). Therefore, we performed an HPTLC analysis because we hypothesized that gangliosides may be related to EMT. All tissues commonly expressed ganglioside GM3; however, ganglioside GD3 was specifically expressed in the CRSwNP_NP tissues (Figure 1B). These findings imply a potential association between ganglioside GD3 and the EMT process.

### 2.2. Inflammation Induced the EMT Process and GD3 Expression upon TNF-α Stimulation

Several studies have demonstrated that CRSwNP is characterized by persistent inflammatory symptoms in the nasal and paranasal mucosa, resulting in the development of NPs [29,30]. Treatment with TNF-α (20 ng/mL) upregulated genes involved in the inflammatory process, including *IL-6*, *IL-8*, and *TNF-α* in hNECs (Figure 2A). Furthermore, an increased inflammatory response was observed based on the increased expression of mesenchymal marker expression (N-cadherin, zinc finger protein SNAI2 [SLUG], and matrix metallopeptidase 9 [MMP-9]), whereas the expression of the epithelial marker E-cadherin was reduced (Figure 2B–D). Notably, the expression of the ganglioside GD3 synthase *ST8SIA1* was increased after TNF-α treatment (Figure 2E); additionally, ganglioside GD3 expression was detected for the first time (Figure 2F). These results indicated that TNF-α treatment induced the inflammatory response, EMT process, and ganglioside GD3 expression.

### 2.3. Direct Correlation between EMT Process and Ganglioside GD3 Expression upon TNF-α Treatment

We found that inflammation induced the EMT process and ganglioside GD3 expression after TNF-α treatment. Therefore, to identify the direct effects of ganglioside GD3 on EMT, we knocked down *ST8SIA1* by transfecting hNECs with three siRNAs targeting *ST8SIA1* (Appendix A). After 48 h, we observed an efficient knockdown of *ST8SIA1* mRNA in hNECs transfected with *siST8SIA1* #1 (Appendix A). *ST8SIA1* knockdown reduced *ST8SIA1* and ganglioside GD3 expression even after TNF-α treatment (Figure 3A,B). Moreover, the inflammatory response (Figure 3C,D), EMT process, and mesenchymal stem cell marker expression (Figure 3E,F) were suppressed when *ST8SIA1*-knockdown hNECs were treated with TNF-α compared with those in negative control transfected hNECs. On the basis of the above observations, we investigated whether the inhibition of wound healing occurred with *ST8SIA1*-knockdown hNECs. Wound healing was suppressed by *ST8SIA1*-knockdown hNECs after TNF-α induction. These results indicated that knockdown of *ST8SIA1* inhibited ganglioside GD3 expression and regulated inflammation, EMT, and wound healing when inflammation was induced by TNF-α.

### 2.4. Inhibition of Ganglioside GD3 Expression Suppressed Inflammation and the EMT Process

Next, to determine whether ganglioside GD3 regulation could attenuate inflammation and the EMT process, we administered NEU3, a mammalian sialidase [31,32], to TNF-α-treated hNECs. The NEU3 treatment decreased *ST8SIA1* and ganglioside GD3 expression in TNF-α-treated cells (Figure 4A,B). The inflammation-related gene expression level and NF-κB signal also decreased after NEU3 treatment in TNF-α-treated hNECs (Figure 4C,D). The promotion of EMT appeared to decrease after the NEU treatment (Figure 4E,F). Moreover, we investigated whether the inhibition of wound healing occurred with NEU3-treated hNECs. Wound healing was suppressed by exogenous NEU3-treated hNECs after TNF-α induction. These data revealed that the inhibition of ganglioside GD3 expression attenuated the inflammatory response, EMT, and wound healing in vitro.

## 3. Discussion

CRS is a heterogeneous and multifactorial condition marked by the participation of various inflammatory cell types [1,2]. Most patients with CRS exhibit activation of biological processes associated with EMT [33]. EMT is closely related to inflammation because it is influenced by the inflammatory response [34,35,36]. EMT is an important mechanism related to tissue remodeling during the development of CRS NPs [13,14,37,38,39]. Furthermore, several investigations have demonstrated the involvement of gangliosides in EMT processes [22,26,40]. Consequently, we obtained sinonasal tissue specimens from patients and then conducted an analysis of both EMT marker and ganglioside expression. Our results indicated that only the CRSwNP tissues expressed N-cadherin and ganglioside GD3. Taken together, we hypothesized that the regulation of ganglioside expression may potentially influence inflammation and the EMT process.

The regulation of inflammation, EMT, and ganglioside expression depends on various extracellular and intercellular factors. Specifically, TNF-α serves as a modulator influencing ganglioside expression and the EMT process [41,42]. TNF-α has been identified as a significant regulator and pathological factor impacting inflammation and autoimmune disease progression [43]. TNF-α stimulation leads to the upregulation of mucin 5AC1; thus, mucin 5AC1 hypersecretion during inflammation plays an important role in the pathogenesis of airway diseases [44,45]. Additionally, TNF-α induces the expression of IL-6 and IL-8 in hNECs [46,47]. Our findings demonstrated that TNF-α treatment induced inflammation in hNECs (Figure 2).

Gangliosides are primarily localized on the external surface of the cell membrane [15,48]. They regulate various biological functions, including cell differentiation, cell-surface interactions, and transmembrane signaling [15,16,17,18,19]. Specifically, ganglioside GD3 is implicated in apoptosis induction, sensitizing tumor cells to anticancer drug therapy, and the suppression of immune system-mediated tumor killing; however, it also suppresses inflammation and cell proliferation [49,50,51,52,53].

Furthermore, TNF-α expression induces ganglioside GD3 expression [41,54,55]. We found that NP tissues alone expressed ganglioside GD3, and TNF-α treatment induced *ST8SIA1* and GD3 expression (Figure 2E,F). Notably, TNF-α plays a crucial role of activating *ST8SIA1* gene expression via NF-κB [56,57,58,59]; therefore, we suppressed *ST8SIA1* gene expression using siRNA. *ST8SIA1*-knockdown hNECs showed significantly reduced *ST8SIA1* gene and GD3 expression as well as suppression of inflammation after TNF-α treatment (Figure 3A–C).

Neuraminidases (NEUs) are a class of glycosidases responsible for cleaving terminal sialic acid residues present on glycoproteins, glycolipids, and oligosaccharides [60]. These NEUs are categorized according to their subcellular locations in lysosomal, cytosolic, and plasma membrane compartments, denoted as NEU1, NEU2, NEU3, and NEU4, respectively [32,60]. Notably, several studies have demonstrated that abnormal ganglioside accumulation triggers inflammation in conditions such as Tay–Sachs disease [61,62]. Therefore, we treated hNECs with NEU3 because gangliosides are abundant and localized on the outer membrane [15,48] and NEU3 is expressed in human airway epithelial cells [63]. We showed that *ST8SIA1* gene and ganglioside GD3 expression were significantly decreased after TNF-α treatment.

Several studies have revealed a relationship between gangliosides and EMT in inflammatory diseases and cancers that exhibit EMT [28,64]. Specifically, ganglioside GD3 synthase expression is necessary in breast cancer for the initiation and maintenance of EMT in TNF-α-induced changes to EMT [26]. We found that ganglioside GD3 is only expressed in CRSwNP_NP tissues (Figure 1B) and TNF-α-treated hNECs (Figure 2F). These results suggest that GD3 may play an important role in polyp formation. Consequently, we regulated GD3 expression using siRNA or NEU3 treatment. Notably, several studies have demonstrated that the inhibition of gangliosides suppresses TNF-α-induced MMP-9 expression [65] and reduces fibrosis [66,67]. In addition, we demonstrated that NEU3-treated hNECs showed reduced expression levels of the EMT markers N-cadherin, SLUG, and MMP-9.

In this study, we report high expression levels of N-cadherin and the detection of ganglioside GD3 expression for the first time in ex vivo CRSwNP_NP tissues. Additionally, we also demonstrated that TNF-α treatment induced inflammation, the EMT process, and ganglioside GD3 expression in hNECs. Furthermore, the regulation of ganglioside expression by siRNA or NEU3 treatment attenuated inflammation and the EMT process. These results suggested the significant role of gangliosides in the inflammatory and EMT processes.

## 4. Materials and Methods

### 4.1. Patients and Tissue Samples

The study protocol was approved by the Institutional Review Board of Konyang University Hospital (KYUH 2019-09-016), and tissues were provided by the bank of human-derived materials from Konyang University Hospital. All research was performed in accordance with the relevant guidelines and regulations. Sinusitis or nasal polyposis was diagnosed in patients with CRS based on personal medical history, physical examination, nasal endoscopy, and computed tomography findings of the sinuses, following the guidelines of the 2012 European position paper on rhinosinusitis and NPs [29]. Exclusion criteria were as follows: (1) age < 18 years; (2) asthma or aspirin sensitivity; (3) use of antibiotics, inhalation of systemic corticosteroids, topical corticosteroids, or other immunomodulatory drugs up to 4 weeks before surgery; and (4) presence of conditions such as unilateral rhinosinusitis, unilateral NPs, antrochoanal polyps, cystic fibrosis, immotile ciliary disease, systemic coagulation disorder, or immunodeficiency. Uncinate process (UP) tissues and NP tissues were collected at the time of endoscopic sinus surgery from patients with CRSwNP or CRSsNP. The patient characteristics are presented in Table 1. UP tissues were obtained from healthy controls (*n* = 5) and patients with CRSsNP (*n*  = 4). Further, NP and UP tissues were obtained from patients with CRSwNP (*n* = 5).

### 4.2. Immunohistochemistry and Immunocytochemistry Analysis

Tissues were fixed in 4% paraformaldehyde (Biosesang, Seongnam, Republic of Korea) in PBS overnight at 4 °C. Fixed tissues were immersed in 30% sucrose overnight at 4 °C and embedded in optimal cutting temperature compound (Sakura Finetek, Torrance, CA, USA). Frozen sections were stained with specific antibodies. For immunocytochemistry, the hNECs were fixed with 4% paraformaldehyde in PBS for 15 min at room temperature (25–28 °C) and permeabilized in 0.25% Triton X-100 for 15 min at room temperature. Cells were blocked with 4% BSA (RMBIO, Missoula, MT, USA) in PBS for 1 h at room temperature and then incubated with specific primary antibodies overnight at 4 °C. After incubation, the cells were washed with 0.05% Tween-20 (Sigma-Aldrich, St. Louis, MO, USA) in PBS and then incubated with Alexa Fluor-conjugated secondary antibodies (Abcam, Cambridge, UK) for 1 h at room temperature. DAPI (Thermo Fisher Scientific, Waltham, MA, USA) was used to stain the nuclei. Fluorescence images were captured using an EVOS M7000 Imaging System (Thermo Fisher Scientific, Waltham, MA, USA). The antibodies used in these experiments are listed in Appendix A.

### 4.3. Extraction and Purification of Gangliosides

The tissues (100 mg/sample) and cells (2 × 10^6^ cells/sample) were homogenized in 0.5 mL of distilled water at 4 °C to extract the total lipids, and then 10 mL of chloroform/methanol (MeOH; 2:1, *v*/*v*) was added to the homogenate. Total lipids were extracted for 30 min at 37 °C, after which the cells were centrifuged at 2000 rpm for 10 min (Hanil Scientific, Gimpo, Republic of Korea). Next, the supernatant was collected, and the pellet was re-extracted twice using the same solvent. The pooled supernatant was then dried at 30 °C in a rotary evaporator (Eyela, Tokyo, Japan), after which the residue was dissolved in 10 mL of chloroform/MeOH/H_2_O (15:30:4, *v*/*v*/*v*) and then loaded onto a DEAE sephadex A-25 column (Sigma-Aldrich, St. Louis, MO, USA) to collect anionic lipid molecules. Next, the column was washed with 10 mL of the same solvent to eliminate neutral lipids, and the adsorbed acidic lipids were eluted using 15 mL of chloroform/MeOH/0.8 M aqueous sodium acetate (15:30:4, *v*/*v*/*v*). The eluted samples were then dried at 30 °C under nitrogen for 5 h, dissolved in chloroform/MeOH (1:1, *v*/*v*), and alkalized in 12 N ammonium hydroxide solution overnight at room temperature. The reaction mixture was evaporated to dryness under nitrogen gas, after which the residue was dissolved in distilled water and applied to a Sep-Pak C18 cartridge (Millipore, Burlington, MA, USA) to eliminate salts. Gangliosides were successively eluted from the cartridge using 2 mL MeOH and 4 mL chloroform/MeOH (2:1, *v*/*v*). The eluted gangliosides were dried at 30 °C under nitrogen for 3 h, and stored at −80 °C until analysis.

### 4.4. High-Performance Thin-Layer Chromatography

High-performance thin-layer chromatography (HPTLC) analysis of the gangliosides was performed using a 10 × 10 cm HPTLC 5651 plate (Merck, Rahway, NJ, USA), as previously described [68]. Purified gangliosides (100 μg of protein/50 μL/lane) were separated using the HPTLC plates, which were subsequently developed using chloroform/MeOH/0.25% CaCl_2_·H_2_O (50:40:10, *v*/*v*/*v*). Gangliosides were visualized via 0.2% resorcinol staining and then the HPTLC plates were dried at 105 °C in a drying oven for at least 3 h. Monosialoganglioside (Matreya LLC, State College, PA, USA), disialoganglioside (Matreya LLC, State College, PA, USA), and rat brain gangliosides were used as markers for individual ganglioside species.

### 4.5. Culture of Human Nasal Epithelial Cells

Primary human nasal epithelial cells (hNECs) from healthy donors were purchased from PromoCell (Heidelberg, Germany). hNECs were cultured in commercially available airway epithelial cell growth medium with supplements (PromoCell, Catalog No. C-21060, Heidelberg, Germany) in a 75 cm^2^ T-flask in a humidified incubator at 37 °C in 5% CO_2_. Cells were treated with 20 ng/mL TNF-α (R&D Systems, Minneapolis, MN, USA) or 50 ng/mL of neuraminidase 3 (NEU3; Origene, Rockville, MD, USA) for 48 h.

### 4.6. Transfection of Short Interfering RNA

hNECs were transfected with short interfering RNA (siRNA) against ST8 alpha-N-acetyl-neuraminide alpha-2,8-sialyltransferase 1 (*siST8SIA1*; Bioneer, Daejeon, Republic of Korea) or non-targeting negative control siRNA (Bioneer, Daejeon, Republic of Korea) using Lipofectamine RNAiMAX Reagent (Thermo Fisher Scientific, Waltham, MA, USA) according to the manufacturer’s protocol. After 24 h, inflammation was induced for 48 h using TNF-α. The siRNA sequences used in this study are listed in Appendix A.

### 4.7. Total RNA Extraction and Quantitative Real-Time PCR

Total RNA was extracted from hNECs using TRIzol reagent (Thermo Fisher Scientific, Waltham, MA, USA), and RNA quality was assessed based on fluorescence using a biospectrometer (Eppendorf, Hamburg, Germany). Reverse transcription was performed using the ReverTra Ace qPCR RT Master Mix with gDNA Remover (Toyobo, Osaka, Japan) according to the manufacturer’s instructions. Real-time quantitative PCR (qPCR) was performed using 2X SYBR Green Master Mix (Thermo Fisher Scientific, Waltham, MA, USA) with gene-specific primers on a 7500 Fast Real-Time PCR System (Applied Biosystems, Waltham, MA, USA). Primer sequences used in this study are listed in Appendix A.

### 4.8. Western Blot Analysis

The cells were lysed using radioimmunoprecipitation assay buffer (Thermo Fisher Scientific, Waltham, MA, USA) containing a protease inhibitor cocktail (Cell Signaling Technology, Danvers, MA, USA) on ice for 30 min and then centrifuged at 20,000× *g* for 30 min at 4 °C. Protein concentrations were determined using a bicinchoninic acid protein assay kit (Thermo Fisher Scientific, Waltham, MA, USA). The proteins in the supernatant were denatured by boiling in SDS sample buffer for 5 min. Total protein (30 μg) from each sample was loaded on an 8% SDS-polyacrylamide gel and electrophoresed at 120 V for 1 h. Separated proteins were transferred to polyvinylidene difluoride membranes (Bio-Rad, Hercules, CA, USA) at 300 A for 1 h. The membranes were blocked using 4% BSA (Bovogen, Melbourne, Australia) at room temperature for 1 h and then incubated with specific primary antibodies overnight at 4 °C. The membranes were washed with Tris-buffered saline containing 0.1% Tween-20 (Sigma-Aldrich, St. Louis, MO, USA) and probed with HRP-conjugated secondary antibodies at room temperature for 1 h. The bands were detected using SuperSignal West Femto Chemiluminescent Substrate (Thermo Fisher Scientific, Waltham, MA, USA) and a LuminoGraph2 imaging system (ATTO, Tokyo, Japan). β-actin was used as an internal control. The antibodies used in these experiments are listed in Appendix A.

### 4.9. Wound Healing Assay

Confluent hNECs monolayers in 6-well plates were wounded by manually scraping the cells with a blue pipette tip. The cells were incubated with or without TNF-α (20 ng/mL). After 24 h, cell migration into the wound surface was captured by microscopy with a camera (Olympus, Tokyo, Japan)

### 4.10. Statistical Analysis

All experiments were repeated at least thrice, and results are presented as the mean ± SD. The unpaired Student’s *t*-test was performed using GraphPad Prism (GraphPad Software, Version 8.00, San Diego, CA, USA).

## Figures and Tables

**Figure 1 ijms-25-04054-f001:**
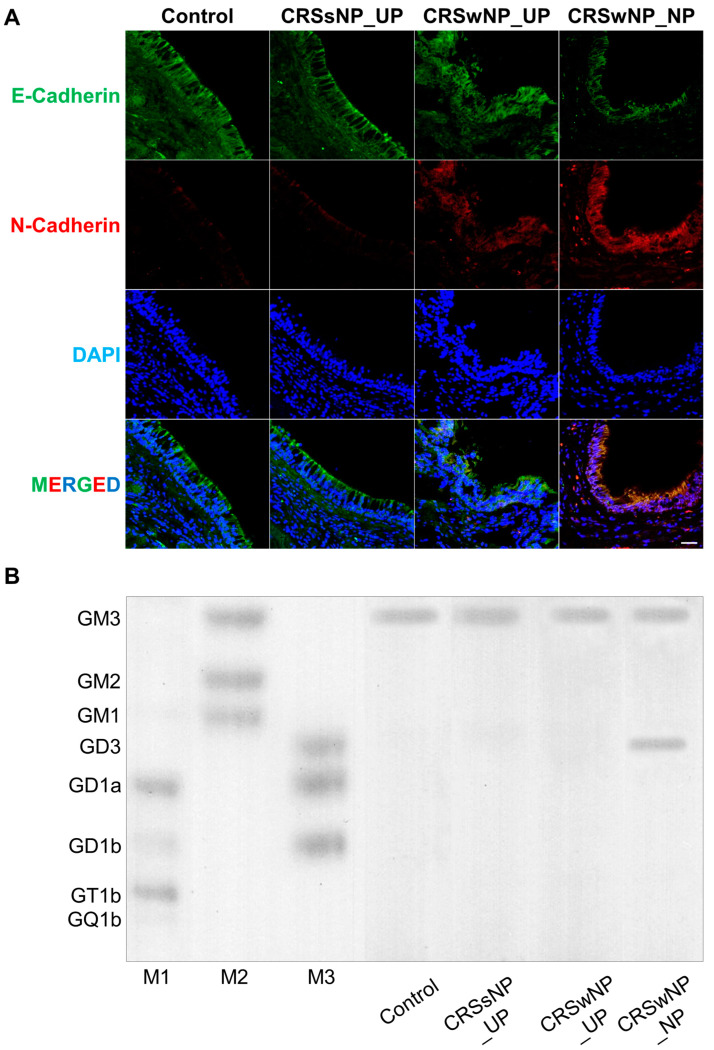
**Expression of epithelial-to-mesenchymal transition markers and gangliosides in patients with CRSwNP.** (**A**) Immunofluorescence staining for E-cadherin (Alexa 488, green) and N-cadherin (Alexa 555, red) within NP and UP tissues obtained from control subjects, patients with CRSsNP, and patients with CRSwNP. DAPI staining was performed to evaluate the nuclei. Scale bar, 500 μm. (**B**) High-performance thin-layer chromatography analysis of ganglioside expression in NP and UP tissues obtained from control subjects, patients with CRSsNP, and patients with CRSwNP. M1, M2, and M3 are ganglioside standard markers. NP, nasal polyp; UP, uncinated process; CRSwNP, chronic rhinosinusitis with nasal polyps; CRSsNP, chronic rhinosinusitis without nasal polyps.

**Figure 2 ijms-25-04054-f002:**
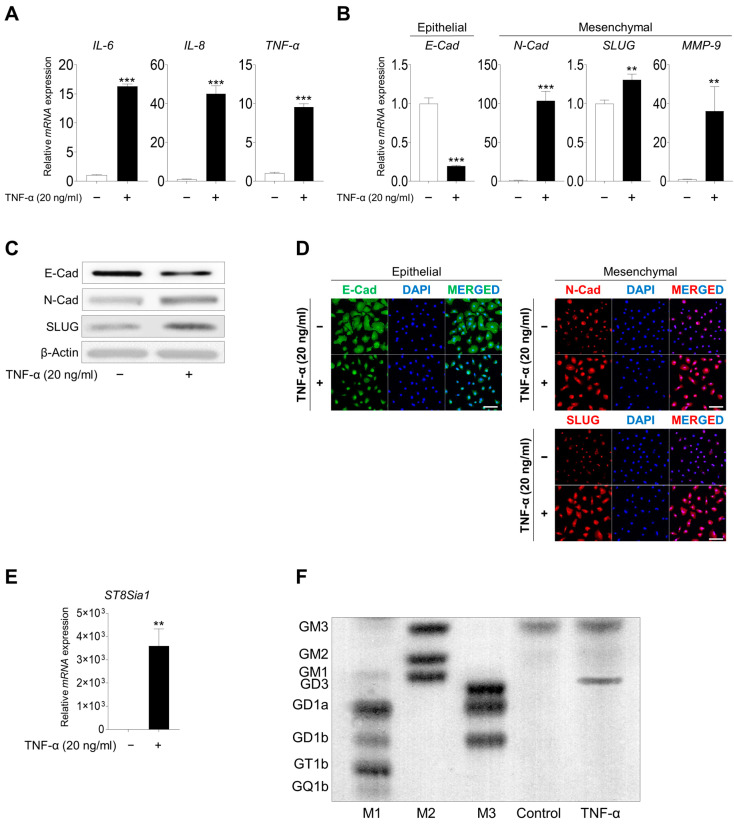
**TNF-α treatment induced the inflammatory response, EMT process, and ganglioside GD3 expression.** (**A**) Expression of inflammation-related genes, IL-6, IL-8, and TNF-α, in hNECs after treatment with TNF-α. (**B**) Comparison of EMT-related gene expression after treatment of hNECs with TNF-α. The mRNA expression levels of epithelial markers (E-cadherin) and mesenchymal markers (N-cadherin, SLUG, and MMP-9) were compared using qPCR. (**C**) Comparison of EMT-related protein expression after treatment of hNECs with TNF-α. Epithelial marker (E-cadherin) and mesenchymal marker (N-cadherin and SLUG) protein expression were compared via Western blotting. (**D**) Immunofluorescence staining of E-cadherin (Alexa 488, green), N-cadherin, and SLUG (Alexa 555, red) in TNF-α-treated hNECs. DAPI staining was used to evaluate the nuclei. Scale bar, 100 μm. (**E**) Comparison of GD3 synthase gene expression after treatment of hNECs with TNF-α. *ST8SIA1* mRNA expression was evaluated using qPCR. mRNA and protein expression levels were normalized to that of the housekeeping gene β-actin. Expression levels are presented as the mean ± SD from three independent experiments and analyzed using Student’s *t*-test. ** *p* < 0.01 and *** *p* < 0.001 compared with non-treated hNECs. (**F**) High-performance thin-layer chromatography analysis of ganglioside expression in TNF-α-treated hNECs. M1, M2, and M3 are ganglioside standard markers. EMT, epithelial-to-mesenchymal transition; hNECs, human nasal epithelial cells; SLUG, zinc finger protein SNAI2; MMP-9, matrix metallopeptidase 9; *ST8SIA1*, ST8 alpha-N-acetyl-neuraminide alpha-2,8-sialyltransferase.

**Figure 3 ijms-25-04054-f003:**
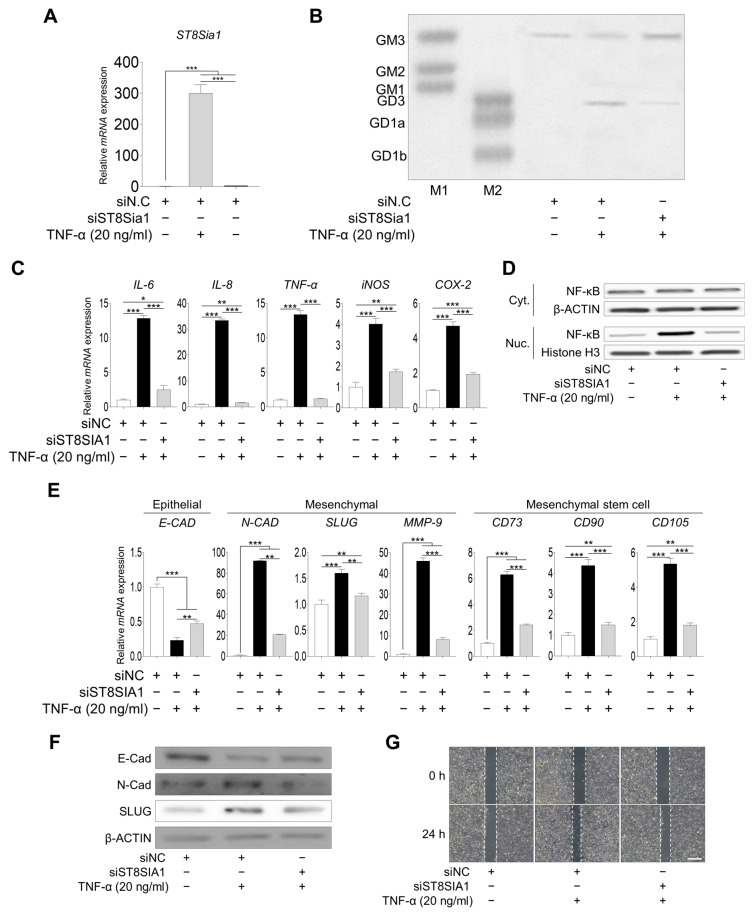
**Change in inflammatory response and EMT process upon *ST8SIA1* knockdown in hNECs.** (**A**) *ST8SIA1* (GD3 synthase) gene expression after treatment of *ST8SIA1*-knockdown hNECs with TNF-α. (**B**) High-performance thin-layer chromatography analysis of ganglioside expression in *ST8SIA1*-knockdown hNECs with or without TNF-α treatment. M1 and M2 are ganglioside standard markers. (**C**) Inflammation-related gene expression after treatment of *ST8SIA1*-knockdown hNECs with TNF-α was determined by qPCR. (**D**) NF-κB/P65 (cytoplasmic and nuclear) were analyzed by Western blotting. EMT-related (**E**) gene and (**F**) protein expression after treatment of *ST8SIA1*-knockdown hNECs with TNF-α. mRNA and protein expression levels were normalized to that of the housekeeping gene β-actin. (**G**) Wound healing assay. ST8SIA1-knockdown hNECs were placed on 6-well plate and wounded by a blue pipette tip and then treated with 20 ng/mL of TNF-α for 24 h. Scale bar 200 μm Expression levels are presented as the mean ± SD from three independent experiments and analyzed using Student’s *t*-test. * *p* < 0.05, ** *p* < 0.01, and *** *p* < 0.001 compared with *siNC* or *siST8SIA1* directly transfected hNECs. EMT, epithelial-to-mesenchymal transition; hNECs, human nasal epithelial cells; *ST8SIA1,* ST8 alpha-N-acetyl-neuraminide alpha-2,8-sialyltransferase; Cyt., cytoplasmic; Nuc., nuclear.

**Figure 4 ijms-25-04054-f004:**
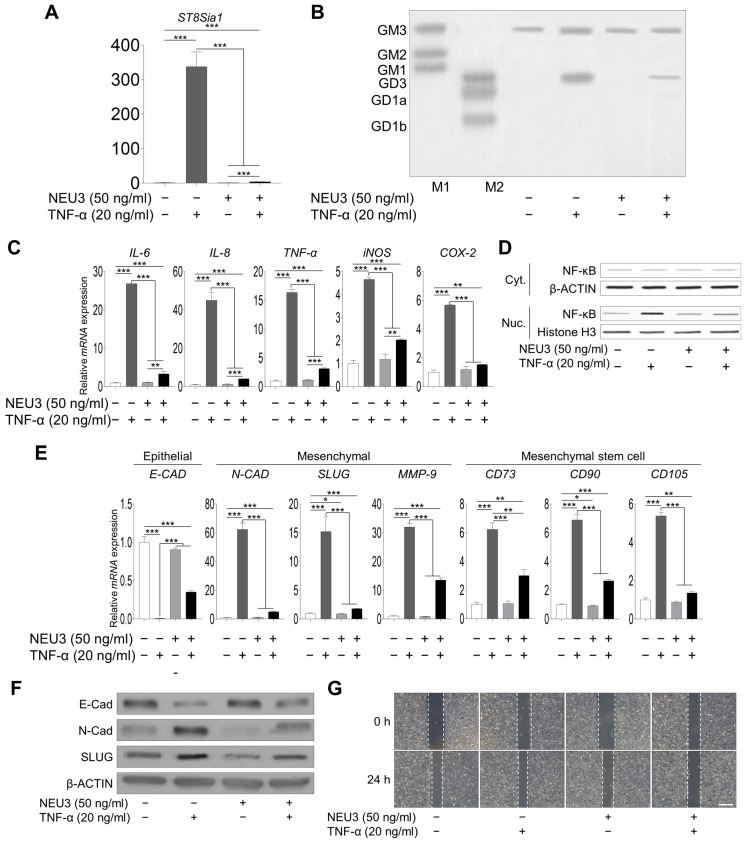
**Effect of treatment with the ganglioside GD3 inhibitor NEU3 on TNF-α-treated hNECs.** (**A**) *ST8SIA1* gene expression in TNF-α-treated hNECs with or without NEU3 treatment. (**B**) High-performance thin-layer chromatography analysis of ganglioside expression in TNF-α-treated hNECs with or without NEU3 treatment. M1 and M2 are ganglioside standard markers. (**C**) mRNA expression levels of inflammation-related genes in TNF-α-treated hNECs with or without NEU3 treatment. (**D**) NF-κB/P65 (cytoplasmic and nuclear) were analyzed by Western blotting. (**E**) mRNA and (**F**) protein expression of epithelial-to-mesenchymal transition-related genes in TNF-α-treated hNECs with or without NEU3 treatment. mRNA and protein expression levels were normalized to that of the housekeeping gene β-actin. (**G**) Wound healing assay. hNECs were placed on 6-well plate and were treated with or without NEU3 (50 ng/mL) for 1 h and wounded by a blue pipette tip. And then treated with 20 ng/mL of TNF-α for 24 h. Scale bar 100 μm. Expression levels are presented as the mean ± SD from three independent experiments and analyzed using Student’s *t*-test. * *p* < 0.05, ** *p* < 0.01, and *** *p* < 0.001 compared with non-treated, TNF-α-treaed or NEU3-treaed hNECs. hNECs, human nasal epithelial cells; *ST8SIA1*, ST8 alpha-N-acetyl-neuraminide alpha-2,8-sialyltransferase; NEU3, neuraminidase 3; Cyt., cytoplasmic; Nuc., nuclear.

**Table 1 ijms-25-04054-t001:** Clinical characteristics of patients included in this study.

Characteristics	Control Patients	Patients with CRSsNP	Patients with CRSwNP	
Subjects (*n*)	8	8	8	8
Tissue used	UP	UP	UP	NP
Age (y)	47.5 ± 15.35	43.06 ± 16.0	46.9 ± 12.52	
Male/Female, *n*/*n*	3/5	5/3	4/4	
Allergic rhinitis (*n*)	0	0	0	
Aspirin sensitivity (*n*)	0	0	0	
Smoking, *n* (%)	1 (12.5%)	2 (25%)	2 (25%)	
Methodologies used				
HPTLC	5	5	5	5
Tissue IHC	3	3	3	3

CRSsNP, chronic rhinosinusitis without nasal polyps; CRSwNP, chronic rhinosinusitis with nasal polyps; UP, uncinate process; NP, nasal polyp.

## Data Availability

Data is contained within the article and Appendix A.

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
