# Peer review of "Ganglioside GD3 Regulates Inflammation and Epithelial-to-Mesenchymal Transition in Human Nasal Epithelial Cells"

_ijms, 2024, doi:10.3390/ijms25074054_

Round 1

Reviewer 1 Report

Comments and Suggestions for Authors

The article titled 'Ganglioside GD3 regulates inflammation and epithelial-to-mesenchymal transition in human nasal epithelial cells' by Ji Hyeon Hwang, Jae-Sung Ryu, Jin Ok Yu, Young-Kug Choo, Jaeku Kang, and Jong-Yeup Kim demonstrates that gangliosides may play important roles in the prevention and treatment of inflammation and EMT in human nasal epithelial cells. The article is well-conceived and organized; however, it could be improved.

1. The authors could improve their investigations by analyzing the NF-κB pathway in hNECs treated with TNF-α (20 ng/ml) and knocked down ST8SIA1.

2. The investigations on epithelial and mesenchymal phenotype (E-cad expression vs. N-CAD and slug) in hNECs treated with TNF-α (20 ng/ml) and knocked down ST8SIA1 could be associated with a wound healing assay. The mesenchymal phenotype could be further enhanced by analyzing the main markers of mesenchymal stem cells (CD73, CD90, CD105).

3. Please improve Figure 3.

4. Please specify the hNECs culture medium in the Materials and Methods section.

5. The discussion is lacking references highlighting the main features of EMT in nasal polyps

Jiang W, Zhou C, Ma C, Cao Y, Hu G, Li H. TGF-β1 induces epithelial-to-mesenchymal transition in chronic rhinosinusitis with nasal polyps through microRNA-182. Asian Pac J Allergy Immunol. 2021 Dec 26. doi: 10.12932/AP-040921-1224. Epub ahead of print. PMID: 34953475.

You B, Zhang T, Zhang W, Pei Y, Huang D, Lei Y, Zhang S, Qiu C, Zhang J, Gu Z, Cheng L, Chen J. IGFBP2 derived from PO-MSCs promote epithelial barrier destruction by activating FAK signaling in nasal polyps. iScience. 2023 Feb 7;26(3):106151. doi: 10.1016/j.isci.2023.106151. PMID: 36866245; PMCID: PMC9972572.

Mesuraca M, Nisticò C, Lombardo N, Piazzetta GL, Lobello N, Chiarella E. Cellular and Biochemical Characterization of Mesenchymal Stem Cells from Killian Nasal Polyp. Int J Mol Sci. 2022 Oct 30;23(21):13214. doi: 10.3390/ijms232113214. PMID: 36362001; PMCID: PMC9656559.

Author Response

We have submitted our response to the reviewer`s comments in a point-by-point.

Reviewer 2 Report

Comments and Suggestions for Authors

This paper brings the results of an elegantly performed study determining the contribution of GD3 to epithelial-to-mesenchymal transition in human nasal epithelial cells. The results are presented clearly and are of interest to the wider audience.

Minor points to consider:

1. There seems to be a mistake in Table 1 where it says Control Patients /   Patients with CRSsNP  /   Patients with CRSsNP. CRSsNP is repeated twice so this needs to be corrected to CRSwNP.

2. In Materials and methods section 4.3. line 279 states “The cells were homogenized…” How many cells? Furthermore, Figure 1B and Results section 2.1. mention ganglioside expression in tissues obtained from control subjects and patients, but the extraction of gangliosides from tissues was not described (tissue wet weight, the means of homogenization, etc). This should be clarified in Materials and methods (section 4.3.)

3. Materials and methods section 4.4. (HPTLC): How much of the extracted gangliosides was spotted on the HPTLC plate? Please add that information.

4. Discussion, line 216 states 3 NEUs: NEU1, NEU2 and NEU3. What about NEU4? It should be added, the authors already cite the reference discussing these 4 NEUSs (reference 33 in the list, Smutova et al)

Author Response

(The authors gave the same response as above.)

Round 2

Reviewer 1 Report

Comments and Suggestions for Authors

The authors significantly improved the manuscript so it is now suitable for pubblication in ijms.